# Direct conversion of lignin to functionalized diaryl ethers via oxidative cross-coupling

Mingyang Liu [1] & Paul J. Dyson [1] ✉

Efficient valorization of lignin, a sustainable source of functionalized aromatic products, would reduce dependence on fossil-derived feedstocks. Oxidative depolymerization is frequently applied to lignin to generate phenolic monomers. However, due to the instability of phenolic intermediates, repolymerization and dearylation reactions lead to low selectivity and product yields. Here, a highly efficient strategy to extract the aromatic monomers from lignin affording functionalized diaryl ethers using oxidative cross-coupling reactions is described, which overcomes the limitations of oxidative methods and affords high-value specialty chemicals. Reaction of phenylboronic acids with lignin converts the reactive phenolic intermediates into stable diaryl ether products in near-theoretical maximum yields (92% for beech lignin and 95% for poplar lignin based on the content of β−O−4 linkages). This strategy suppresses side reactions typically encountered in oxidative depolymerization reactions of lignin and provides a new approach for the direct transformation of lignin into valuable functionalized diaryl ethers, including key intermediates in pharmaceutical and natural product synthesis.

Among the plentiful biomass, lignin is the only sustainable source of aromatic compounds[1–6] and is available in abundant quantities as a waste product from the pulp and paper and bioethanol industries[6]. Nonetheless, lignin is under-exploited as a renewable chemical feedstock due to the limited number of efficient and selective downstream processing strategies available.

Various methods have been extensively studied for the conversion of lignin into aromatic products that can broadly be classified as catalytic oxidative degradation, catalytic reductive degradation and acid/base-catalyzed degradation[2–4,6]. Catalytic oxidative degradation has a number of advantages compared to other catalytic fractionation methods including hydrogenolysis (a reductive process), and acid/base-catalyzed degradation[4,7,8]. Catalytic oxidative degradation advantageously take place under mild and environmentally benign conditions, which contrasts with hydrogenolysis that uses noble metal catalysts, high reaction temperatures and pressures, or the use of corrosive acid/base-catalyzed degradation reagents. In addition, catalytic oxidative degradation has the potential to retain key functionality in the products that could be relevant in subsequent synthetic

steps[9–11]. Many homogeneous and heterogeneous catalytic oxidative methods that cleave the C−C bonds of the alkyl side chains to depolymerize lignin have been reported, but typically they are limited by poor selectivity and consequently low product yields (Fig. 1)[12–23]. The catalysts reported for the oxidative fragmentation of lignin are summarized in Supplementary Table S1.The critical issue that must be solved to overcome the aforementioned limitations is that the phenolic hydroxy group is unstable under oxidative conditions[24], leading to side reactions including repolymerization and ring opening reactions, which generates complex polymers, oligomers, and non-aromatic side products[25–28]. Hence, aromatic products are isolated in low yields in certain direct oxidative degradation reactions.

Diaryl ethers are typically prepared from cross-coupling reactions between petrochemical-derived substrates, specifically, phenols with excess electrophiles, i.e. aryl halides, or nucleophiles, i.e. boronic acids[29]. Inspired by Cu-catalyzed oxidative cross-coupling reactions between of nucleophiles (phenols, anilines, etc.) with organoboronic acids to afford poly-functionalized diaryl ethers and diaryl amines[30–34], we decided to explore the utility of organoboronic acids to extract the

[1]Institute of Chemical Sciences and Engineering, Ecole Polytechnique Fédérale de Lausanne (EPFL), 1015 Lausanne, Switzerland.
✉e-mail: paul.dyson@epfl.ch

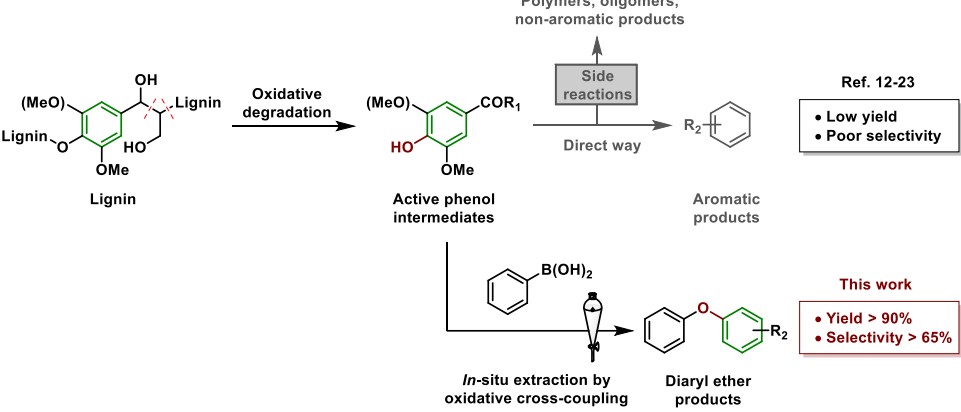

**Fig. 1 | Current catalytic oxidative degradation approach and the approach disclosed herein.** COR₁: −CHO or −COOH. R₂: −H, −OMe, −CHO or −COOH.

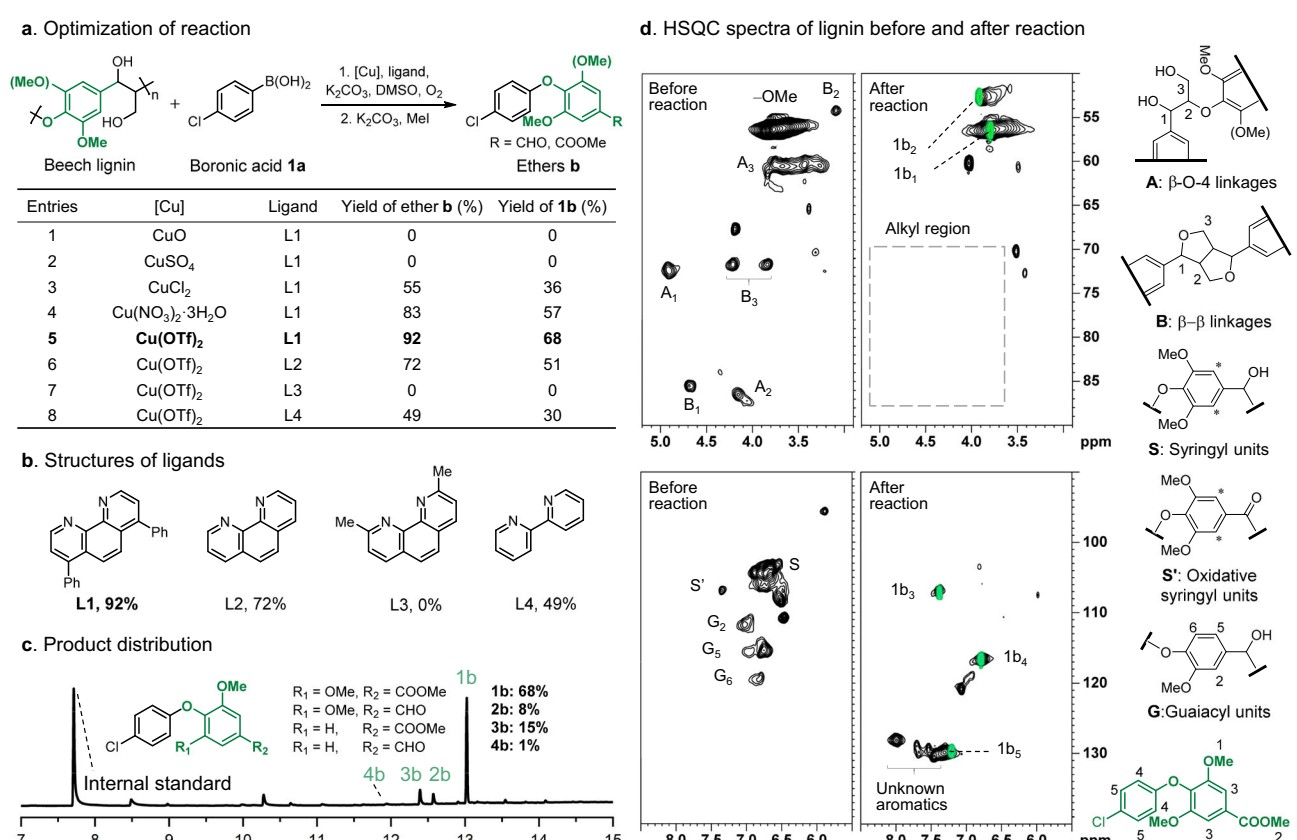

**Fig. 2 | Reaction of beech lignin with 4-chlorophenylboronic acid (1a).**
**a** Reaction and optimization of the catalyst and co-catalyst, further details of the optimization of the reaction parameters are provided in the SI. Reaction conditions: (1) beech lignin (40 mg), 4-chlorophenylboronic acid (1.5 equiv.), Cu salt (30 mol%), co-catalyst (30 mol%), K₂CO₃ (4 equiv.), biphenyl (0.02 mmol, internal standard), DMSO (2 mL), O₂ (3 atm), 140 °C, 6 h. (2) K₂CO₃ (3 equiv.), MeI (10 equiv.), 25 °C, 12 h. The methylation step is required so the products can be

analyzed by GC-MS. Yield of diaryl esters is based on the content of β−O−4 ether linkages in lignin. **b** Structures of co-catalysts. **c** GC spectrum of the reaction mixture (Entry 5, Fig. 2a) showing the presence of four products and their relative abundance. **d** Short-range HSQC spectra before and after reaction. Assignment of contours is provided by the numbering in the structures on the right. Overlying green contours in the HSQC spectra correspond to **1b**.

reactive phenolic intermediates generated during the oxidative degradation of lignin (Fig. 1). We discovered that this protocol prevents common side reactions initialed by phenolic intermediates allowing functionalized diaryl ethers to be obtained in near-theoretical maximum yields. Using lignin as a starting material to synthesize functionalized diaryl ether is advantageous as lignin is an abundant, inexpensive and renewable material[1–6]. The direct conversion of lignin to diaryl ethers in a single step process requires fewer reagents and

solvents than a two-step process in which phenols are initially generated from lignin and then further transformed[5,10,23].

## Results and discussion

Initially, organosolv beech lignin and 4-chlorophenylboronic acid (**1a**) were reacted with O₂ in the presence of various Cu salts in a weakly alkaline solution that mimic the conditions typical of coupling[30–34] and C−C bond cleavage reactions[12,13,15,18] (Fig. 2a). After the oxidative

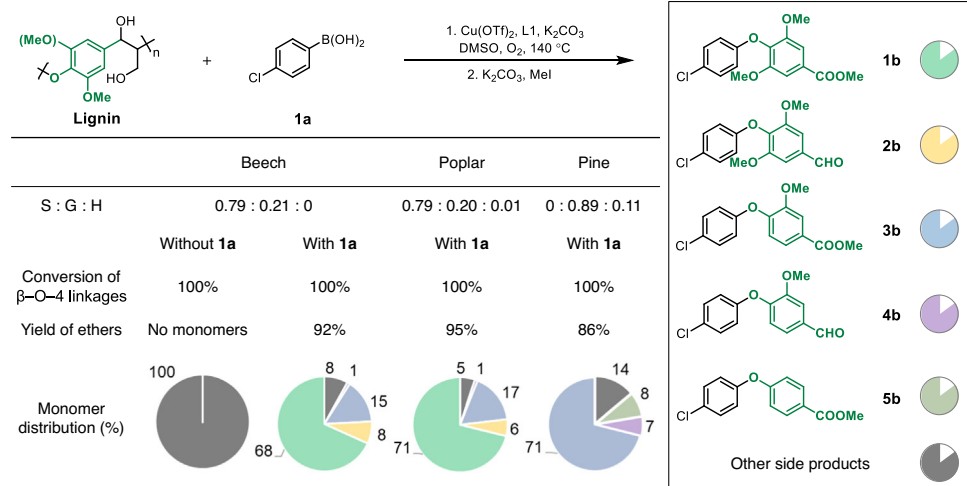

**Fig. 3 | Aromatic ethers generated from lignin using different types of wood.** Other side products corresponds to polymers, oligomers, and non-aromatic side products[25–28]. Standard reaction conditions: (1) Lignin (40 mg), 4-chlorophenylboronic acid (1.5 equiv.), Cu(OTf)₂ (30 mol%), L1 (30 mol%), K₂CO₃ (4 equiv.), biphenyl (0.02 mmol, internal standard), DMSO (2 mL), O₂ (3 atm), 140 °C, 6 h. (2) K₂CO₃ (3 equiv.), MeI (10 equiv.), 25 °C, 12 h.

degradation/coupling step, a methylation step was performed to transforms any carboxylic acid groups into methyl ester groups, to facilitate gas chromatography (GC) analysis. From the Cu salts screened, copper(II) triflate (Cu(OTf)₂) was found to be the most effective catalyst together with bathophenanthroline (L1 in Fig. 2b) co-catalyst, affording the diaryl ether products (**1-4b**) in 92% yield (Fig. 2a, Entry 5). Note that other ligands were evaluated (Fig. 2a, b), but none were as effective as L1. Since the triflate anions in Cu(OTf)₂ are weakly coordinating and readily displaced[35,36], Cu(OTf)₂ is expected to react with L1, a bidentate N-donor ligand, to form the active catalyst in situ. L1 is an electron-rich ligand that is expected to increase the electron density on the Cu center, which facilitates oxidation Cu(II) to Cu(III), a key step in the reaction (the more electron rich the Cu center the easier it is to oxidize)[31,37]. A CuL1 complex is formed in situ (evidenced by mass spectrometry, Supplementary Fig. 1), which serves as the actual catalyst, and is sufficiently stable to be isolated after reaction, recycled and reused with only a minor loss in activity (Supplementary Fig. 1). The GC spectrum showing the product distribution is given in Fig. 2c and Supplementary Fig. 2. As a representative hardwood[2], the aromatic rings of beech wood consist of 79.5% syringyl (S), and 20.5% guaiacyl (G) units (Supplementary Fig. 3). The main diaryl ether products are the syringyl type methyl ester (**1b**, 68%) and the guaiacyl type methyl ester (**3b**, 15%), along with syringyl- and guaiacyl-type aldehyde products (**2b**, 8% and **4b**, 1%, respectively), demonstrating that the aromatic rings of lignin may be extracted successfully as functionalized aromatic diaryl ether products using this method. Detail optimization of the reaction parameters is summarized in Supplementary Tables 2–11.

The evolution of the reaction was monitored by short-range ¹³C-¹H correlation (HSQC) NMR spectroscopy (Fig. 2d shows the spectra before and after reaction). The beech lignin structure consists of syringyl and guaiacyl aromatic rings, together with their main alkyl side chains (β−O−4 (A) and β−β (B) linkages)[2], clearly identified in the HSQC spectrum before reaction. After reaction, signals corresponding to the alkyl side chains and electron-rich aromatic rings are no longer present, indicative of complete degradation of the lignin structure. The HSQC spectrum of the reaction mixture is consistent with that of the expected major product **1b** (Fig. 2d, green contour) and other products, based on comparisons with full HSQC spectra of **1b**–**4b** (Supplementary Figs. 7–10). The remaining HSQC signals may be attributed to unreacted linkages of lignin and side products derived from the excess boronic acid used (Supplementary Fig. 2).

In the absence of **1a**, no aromatic monomer products were detected under the optimized reaction conditions, confirming that the boronic acid captures the reactive phenolic intermediates (Fig. 3). The composition and structure of lignin varies with the type of wood used (Supplementary Figs. 3–5), hence several types were evaluated (Fig. 3). Lignin from poplar, a hardwood, is primarily composed of S aromatic units. Poplar lignin was evaluated under the optimized reaction conditions, affording diaryl ethers in 95% yield, with high selectivity to the syringyl-type product **1b**, which was obtained in 71% yield (Fig. 3). Pine lignin, containing G and coumaryl (H) aromatic units without S units, affords the G-type diaryl ether **3b** as main product in 71% yield (Fig. 3). Raw beech wood sawdust was tested using the standard reaction conditions with diaryl ether products obtained in only 10% yield (Supplementary Fig. 11).

The scope of the boronic acid and related coupling reagents was also evaluated using beech wood lignin under the optimized reaction conditions (Fig. 4). Specifically, phenylboronic acid (**2a**) affords the S type diaryl ether (**6b**) in 66% yield. Phenylboronic acids functionalized with electron-withdrawing halogen (**1a, 3-4a**) or trifluoromethyl (**5-6a**) substituents are tolerated and afford the desired diaryl ether products in 64−71% yield. Nitro (**7a**), ester (**8a**), and methoxy (**9a**) substituents at the *para*-position result in slightly lower yields (49−56%). Biphenyl boronic acid (**10a**) was transformed in 61% yield. Several organic borate esters (**11a**–**13a**) were also tested and are less effective in the aromatic extraction reaction (yields ranging from 23 to 55%), and the phenyltrifluoroborate salt (**14a**) and alkyl boronic acid (**15a, 16a**) do not function as coupling reagents. In general, phenylboronic acids and borate esters could be employed as effective extraction reagents for the transformation of lignin into functionalized diaryl ether products. Diaryl ethers with *ortho*-methoxy and/or *para*-carboxy groups are commonly encountered as intermediates in the preparation of pharmaceutical and natural products, such as chrysophaentins[38,39], himalain A[40] and certain biological inhibitors[41,42].

Mechanistic studies were conducted to probe the reaction pathway and based on the product analysis (Figs. 2d and 3), the transformation of lignin to diaryl ether products appears to involve well-ordered C−C and C−O bond cleavage to release reactive phenol intermediates, which subsequently undergo Cham-Lam coupling with the boronic acid. Control experiments employing diol (**c**, 1-phenyl-1,2-propanediol), benzyl alcohol (**d**), and acetophenone (**e**) as substrates were conducted to confirm that C_α−C_β bond cleavage takes place under the reaction conditions (Fig. 5a). All three different kinds of

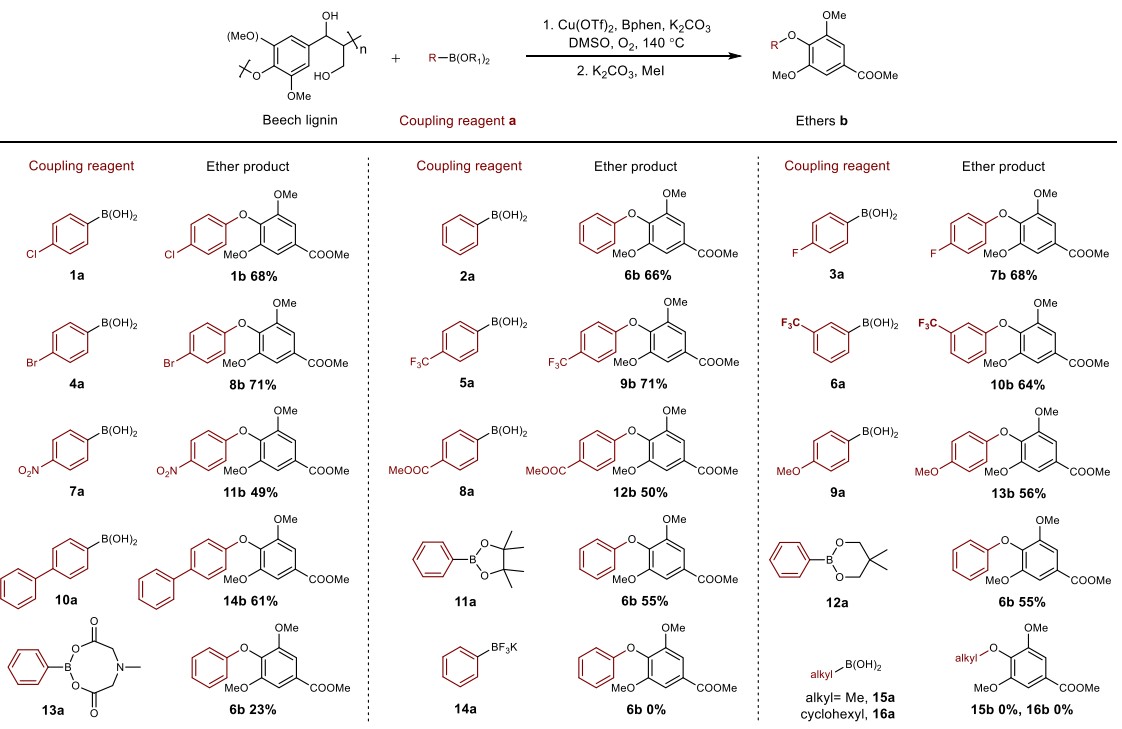

**Fig. 4 | Substrate scope of the boronic acid and related coupling reagents.**
Reaction conditions: (1) Beech lignin (40 mg), boronic acid or borate ester
(1.5 equiv.), Cu(OTf)₂ (30 mol%), L1 (30 mol%), K₂CO₃ (4 equiv.), biphenyl
(0.02 mmol), DMSO (2 mL), O₂ (3 atm), 140 °C, 6 h. (2) K₂CO₃ (3 equiv.), MeI (10
equiv.), 25 °C, 12 h.

$C_\alpha$–$C_\beta$ bonds were successfully cleaved affording methyl benzoate in
excellent yield (90−98%). These results are in agreement with previous
reports concerning Cu catalyzed C−C activation initiated by the oxi-
dation of hydroxy groups (Supplementary Figs. 12 and 13)[13,15,18,43,44]. In
addition, the reaction of protected phenols in which the hydroxy
group is modified with glycol (**f**), acetic acid (**g**) and formic acid (**h**)
groups with phenyl boronic acid (**2a**) affords the expected diphenyl
ether product in 97−99% yield (Fig. 5b). Hence, it would appear that
under the oxidative conditions the protective groups are unstable[15],
and release reactive phenolic intermediates (Supplementary Fig. 14),
that react with the boronic acid.

Aromatic dimers with alkyl β−O−4 linkages were employed as
lignin model compounds and reacted with **2a** under the standard
conditions (Fig. 5c). Dimer **i** containing a stable methoxy group at the
*para* position provides ester **j** and diaryl ether **16b** in good yield,
demonstrating the ability of the catalytic system to degrade the alkyl
β−O−4 linkage via C−C and C−O bond cleavage. Alternatively, with a
hydroxy group at the *para* position (dimer **k**) diaryl ethers **17b** and **16b**
are obtained, which implies the boronic acid reacts directly with the
phenolic hydroxy group.

The roles of Cu(OTf)₂, L1 and K₂CO₃ were investigated through a
series of control experiments (Supplementary Fig. 15). The Cu complex
and base are indispensable for aerobic −OH group oxidation, C−C
bond activation and Cham-Lam coupling. L1 coordinated to the Cu
center to promote the Cham-Lam coupling of the boronic acid with the
phenol intermediates. The kinetic study demonstrates that C−C bond
activation to release reactive phenolic intermediates is slower Cham-
Lam coupling between phenol and boronic acid, which ensures rapid
capture of phenolic intermediates with boronic acid, preventing side
reactions (Supplementary Fig. 16).

Based on the mechanistic studies a tentative reaction pathway is
proposed (Fig. 5d). Under the basic reaction conditions lignin **1** is
activated by the Cu catalyzed oxidation of the hydroxy groups to give
intermediate **2** containing a carbonyl group (Supplementary

Fig. 12)[45,46]. Cu catalyzed $C_\alpha$–$C_\beta$ bonds cleavage of **2** results in the
depolymerization of lignin to the oxalic acid protected phenol
monomer **3** (Supplementary Fig. 13)[13,18,43,44]. Thermal decomposition of
**3** would release the phenolic hydroxy group to generate reactive
intermediate **4** (Supplementary Fig. 14)[47–49], which is captured by
Cham-Lam coupling with an appropriate boronic acid to afford the
carboxylic acid containing product **5**[50,51]. Methylation of the carboxylic
acid in **5** to ester **6** is performed to facilitate analysis.

We developed an *in-situ* method to extract the aromatic rings in
lignin by adding aryl boronic acids or borate esters during oxidative
decomposition. Diaryl ether products are generated in near-
theoretical maximum yield as repolymerization and dearylation side
reactions of the reactive phenol intermediates are suppressed. Studies
indicate that the reaction is initiated by Cu catalyzed C−C and C−O
bond cleavage of the alkyl side chains, degrading the polymeric
structure of lignin and releasing reactive phenol intermediates, which
are captured by cross-coupling with the aryl boronic acids or borate
esters to afford stable diaryl ether products. This study paves the way
to alternative approaches to transform lignin via reactive phenol
intermediates into high-value specialty chemicals, including pharma-
ceutical and natural product intermediates and other chemicals.

## Methods

### Typical reaction to prepare ether monomers from lignin

Beech lignin (40 mg, content of β−O−4 linkages is 0.245 mmol/g
lignin), 4-chlorophenylboronic acid (1.5 equiv.), Cu(OTf)₂ (30 mol%),
bathophenanthroline (L1, 30 mol%), K₂CO₃ (4 equiv.), biphenyl
(internal standard, 2 equiv.), and DMSO (2 mL) were added into
autoclave. The reactor was heated to 140 °C for 6 h. After cooling to
room temperature, the crude product was methylated by addition of
K₂CO₃ (3 equiv.), MeI (10 equiv.) and stirring for 12 h. After reaction,
the mixture was added ethyl acetate (EA, 3 mL) and a saturated NH₄Cl
(5 mL) aqueous solution was added to extract the organic products.
The aqueous solution was extracted into EA (3 × 3 mL). For qualitative

**Fig. 5 | Mechanistic studies. a** Control experiments probing $C_\alpha$–$C_\beta$ bond cleavage. Reaction conditions are the same as in Fig. 4, but without boronic acid. **b** Control experiments to investigate C(alkyl)−O bond cleavage. Reaction conditions are the same as Fig. 4. **c** Exploration of C–C and C–O bond cleavage in β–O–4 lignin model compounds. Reaction conditions were the same as Fig. 4. **d** Plausible reaction mechanism.

and quantitative analysis of the products, the combined organic phase was filtered over silica gel to remove the inorganic salts and ligand. Further purification was achieved using column chromatography.

## Data availability
All data supporting this study are available in the article and Supplementary Information files, and also are available from the corresponding author if request.

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

## Acknowledgements

This publication was supported by EPFL and was created was created as part of NCCR Catalysis (Grant number 180544, Paul Dyson), a National Centre of Competence in Research funded by the Swiss National Science Foundation. We also thank Dr. Manli Hua for helpful discussions.

## Author contributions

M.L. and P.D. designed the research, analyzed data, and wrote the paper. M.L. performed the research.

## Competing interests

The catalytic method is described in a patent invented by M.L. and P.D. (EP Patent, No. 22178410.1).
