## [Peer Review File · Nature Communications]

REVIEWER COMMENTS

Reviewer #1 (Remarks to the Author):

This is an interesting paper that describes an oxidative fragmentation of Lignin and a subsequent cross-coupling reaction of active phenolic intermediates with 4-chlorophenylboronic acid using Cu complex catalysts. Authors have already studied a series of Cu catalysts to synthesize methyl esters with Lignin as a source of “methylating reagents”, in which CO was simultaneously formed as a co-product (Angew Chem. Int. Ed., 2022, 61, e202209093). This time authors found out the combination of Cu catalyst (Copper triflate) and L1 (Bphen) ligand to be effective for the current reaction system. The high yields of diaryl ether products obtained in this reaction system are quite attractive, but novelty and significant advance of this work is weak in terms of “organic chemistry” and “catalytic chemistry”.

1. Lignin is recognized as a raw material to produce “bulk chemicals (aromatics, phenols, etc)”. Authors had better rationalize the motivation and social impact to synthesize “specialty chemicals” using large amounts of stoichiometric reagents and excess 4-chlorophenylboronic acid.
2. Introduction should contain comprehensive overview using homogeneous and heterogeneous catalysts in oxidative fragmentation of Lignin with literature survey. A table in the supporting information summarizing the results of previous papers will be helpful for readers to understand the background of this work.
3. What is the reason to show high activity of Cu(OTf)₂-L1 for this system, compared to others? What is the role of Cu(OTf)₂ and L1 ligand? What is the reason to suppress “side reactions”? Is it possible to explain side reactions to be suppressed with kinetics studies?
4. How about reusability of catalyst and Ligand? How about TOFs or TONs?

Reviewer #2 (Remarks to the Author):

The manuscript “Direct Conversion of Lignin to Functionalized Diaryl Ethers via Oxidative Cross-coupling” by Prof. Paul J. Dyson and Mingyang Liu describes a novel strategy of phenolic monomers' extraction from lignin. In this study, the authors applied oxidative copper-catalyzed cross-coupling reaction of beech lignin with phenylboronic acids affording functionalized diaryl ethers in a simple and highly efficient manner.

The authors proposed an effective procedure of lignin valorization consisting of two steps: oxidative coupling and methylation of the acid. The reaction conditions were optimized, including copper (II) source, base, ligand, solvent, catalyst loading, oxygen pressure, time, and temperature. As one of the most significant advances established by the authors, this protocol enables an efficient generation of various ethers from beech, poplar, and pine lignin in near-theoretical maximum yields with suppression of repolymerization and dearylation side reactions of the reactive phenol intermediates. The authors provided a detailed analysis of the HSQC spectra of the initial lignin polymer and final reaction mixtures. The authors found a correlation between the ratio of the linkages in lignin of three different types and the structures of the observed products. A scope of arylboronic acids with various substituents (EDG and EWG) was performed. Finally, several convincing mechanistic studies were elaborated: cleavage of C-C and C-O bonds and Cham-Lam coupling with simple substrates *c-i*, *k* under standard conditions. For the technical part, the supporting information includes the perfectly elaborated calculation of the ethers' yield based on starting lignin, as well as the linkages content calculation and calibration by the HSQC NMR spectroscopy technique. The spectral data corresponds to the proposed structures and is well-delivered. The citations are perfectly balanced.

Overall, I believe this impactful discovery dramatically pushed the boundaries of the lignin depolymerization topic. Undoubtedly, these advances in the direct conversion of lignin will interest a broad scientific audience. Thus, I fully support the publication of this manuscript in *Nat. Commun.* with minor revisions and four additional experiments outlined below.

- 1) Have you tried to subject lignocellulosic biomass, e.g., wood sawdust, directly to your reaction conditions? Is it possible to isolate any quantity of the pure desired product 1b from lignocellulose with the proposed method?
- 2) Does coupling of beech lignin with cyclohexylboronic acid (CAS 4441-56-9) result in product 1b under the established conditions? Although there is an entry with methylboronic acid 15a, a cyclic alkylboronic acid should be tested.
- 3) Can you recycle copper (II) heterogeneous catalyst and reuse it? How efficient will the recycled catalyst be for beech lignin valorization to produce 1b?
- 4) What is the product yield with 10 mol% amount of Cu(II) catalyst?
- 5) Although poplar lignin is a bit more efficient under your reaction conditions (95% yield of ethers, 71% of 1b), you performed a scope with beech lignin (92% yield of ethers, 68% of 1b). Is there any explanation for this choice?
- 6) In the manuscript, Figure 2a: the heading of the table 'Yield of ether (%)' should be corrected to "Yield of ethers b (%)" for clarity.
- 7) In the manuscript, line 63: specification should be added: "the diaryl ether products (1-4b) in 92% yield (Fig. 2a, Entry 5).
- 8) In the manuscript, a general comment: please enhance the quality of the chemical structures in the schemes.
- 9) In the SI, line 221: rename to Figure S12.

Reviewer #1 (Remarks to the Author):

This is an interesting paper that describes an oxidative fragmentation of Lignin and a subsequent cross-coupling reaction of active phenolic intermediates with 4-chlorophenylboronic acid using Cu complex catalysts. Authors have already studied a series of Cu catalysts to synthesize methyl esters with Lignin as a source of “methylating reagents”, in which CO was simultaneously formed as a co-product (Angew Chem. Int. Ed., 2022, 61, e202209093). This time authors found out the combination of Cu catalyst (Copper triflate) and L1 (Bphen) ligand to be effective for the current reaction system. The high yields of diaryl ether products obtained in this reaction system are quite attractive, but novelty and significant advance of this work is weak in terms of “organic chemistry” and “catalytic chemistry”.

Thank you for these positive remarks, indeed the focus of this work is on sustainable chemistry.

1. Lignin is recognized as a raw material to produce “bulk chemicals (aromatics, phenols, etc)”. Authors had better rationalize the motivation and social impact to synthesize “specialty chemicals” using large amounts of stoichiometric reagents and excess 4-chlorophenylboronic acid.

We have expanded the introduction to better justify the approach used:

Diaryl ethers are typically prepared from cross-coupling reactions between petrochemical-derived substrates, specifically, phenols with excess electrophiles, i.e. aryl halides, or nucleophiles, i.e. boronic acids.³⁴ Using lignin as a starting material to synthesize functionalized diaryl ether is advantageous as lignin is an abundant, inexpensive and renewable material.¹⁻⁶ The direct conversion of lignin to diaryl ethers in a single step process requires fewer reagents and solvents than a two-step process in which phenols are initially generated from lignin and then further transformed.^{5, 10, 23}

2. Introduction should contain comprehensive overview using homogeneous and heterogeneous catalysts in oxidative fragmentation of Lignin with literature survey. A table in the supporting information summarizing the results of previous papers will be helpful for readers to understand the background of this work.

We have expanded the discussion as requested and prepared and added a table to the SI:

Many homogeneous and heterogeneous catalytic oxidative methods that cleave the C–C bonds of the alkyl side chains to depolymerize lignin have been reported, but typically they are limited by poor selectivity and consequently low product yields (Fig. 1).¹²⁻²³ The catalysts reported for the oxidative fragmentation of lignin are summarized in Table S1.

Table S1. Catalysts used for the oxidative fragmentation lignin valorization.

Year	Catalyst	Source	T (°C)	Gas	Product(s)	Yield (%)	Mechanistic study	Ref.
Homogeneous systems								
2014	CuSO ₄ /phen	Pine MW lignin	80	12 bar O ₂	Vanillin, vanillic acid	<10	No	7
2016	Peracetic Acid	Pretreated lignin	60		Phenol mixtures	22	Yes	8
2017	CuSO ₄	Pine wood	170	3 bar O ₂	Vanillin	18	No	9

2018	NaOH	Poplar lignin	175	5 bar O ₂ + 15 bar He	Phenol and carboxylic acid mixtures	30	No	10
2020	NaOH	Wood sawdust	160	10 bar O ₂	Phenol and carboxylic acid mixtures	20	No	11
2021	Bobbitt's salt	Lignin fraction	100		2,6-dimethoxybenzoquinone	18	Yes	12
Heterogeneous systems								
2018	Au/Li–Al layered double hydroxide	Organosolv lignin	120	1 bar O ₂	Phenol and carboxylic acid mixtures	40	Yes	13
2019	Polyoxometalate ionic liquid [BSmim]CuPW ₁₂ O ₄₀	Lignin	170	8 bar O ₂	Diethyl Maleate	404.8 mg/g	No	14
2019	Polyoxometalate K ₅ V ₃ W ₃ O ₁₉	Lignin	115	50 bar O ₂	Carboxylic acids	<20	No	15
2021	Co-N-C catalyst	Poplar biomass	190	35 bar 6% O ₂ in N ₂	Phenol mixtures	15	No	16
2021	Polyoxometalate	Wood sawdust	140	10 bar O ₂ in N ₂ (9:1)	Phenol mixtures	46	No	17

3. What is the reason to show high activity of Cu(OTf)₂-L1 for this system, compared to others? What is the role of Cu(OTf)₂ and L1 ligand? What is the reason to suppress “side reactions”? Is it possible to explain side reactions to be suppressed with kinetics studies?

We added further text to explain the high activity of Cu(OTf)₂-L1:

Since the triflate anions in Cu(OTf)₂ are weakly coordinating and readily displaced,³⁵⁻³⁶ Cu(OTf)₂ is expected to react with L1, a bidentate N-donor ligand, to form the active catalyst in situ. L1 is an electron-rich ligand that increases the electron density on the Cu center, which facilitates oxidation Cu(II) to Cu(III), a key step in the reaction (i.e. the more electron rich the Cu center the easier it is to oxidize).^{30, 37}

Further details to demonstrate the role of catalyst and ligand based on a series of additional control experiments are provided in the SI:

The roles of Cu(OTf)₂, L1 and K₂CO₃ were investigated through a series of control experiments (Fig. S15). The Cu complex and base are indispensable for aerobic –OH group oxidation, C–C bond activation and Cham-Lam coupling. L1 coordinated to the Cu center to promote the Cham-Lam coupling of the boronic acid with the phenol intermediates.

Figure S15. Control experiments. (a) Catalytic oxidation of 1,1-diphenylmethanol **n** to benzophenone **o**. (b) Catalytic C–C bond activation of acetophenone **p** to methyl benzoate **m** and benzaldehyde **q**. (c) Catalytic cross coupling of phenylboronic acid **2a** and 2,6-dimethylphenol **r**. (d) Summary of the functions of Cu(OTf)₂, L1 and K₂CO₃.

We have performed a kinetic study that helps to rationalize the suppression of side reactions:

The kinetic study demonstrates that C–C bond activation to release reactive phenolic intermediates is slower than Cham-Lam coupling between phenol and boronic acid, which ensures rapid capture of the phenolic intermediates with boronic acid, preventing side reactions (Fig. S16).

Figure S16. Kinetic study. Reaction 1 corresponds to the cross-coupling reaction between boronic acid **2a** and lignin model compound **1** to afford diaryl ether **16b** and byproduct **m** (combination of -OH group oxidation, C-C bond activation and Cham-Lam coupling). Reaction 2 is the oxidation of 1,1-diphenylmethanol **n** to benzophenone **o** (-OH group oxidation). Reaction 3 is the oxidation of acetophenone **p** to methyl benzoate **m** and benzaldehyde **q** (C-C bond activation). Reaction conditions are the same as those reported in Fig. S15. Reaction 4 corresponds to the cross-coupling of phenylboronic acid **2a** and 2,6-dimethylphenol **r** to afford diaryl ether **16b** (Cham-Lam coupling). The order of reaction rate is: reaction 2 > reaction 4 > reaction 3 \approx reaction 1.

4. How about reusability of catalyst and ligand? How about TOFs or TONs?

We performed recycling experiments and added the following discussion:

A CuL1 complex is formed in situ (evidenced by mass spectrometry, Fig. S1), which serves as the actual catalyst, and is sufficiently stable to be isolated after reaction, recycled and reused with only a minor loss in activity (Fig. S1).

Figure S1. Isolation and reuse of the CuL1 complex. (a) ¹H NMR spectra of the isolated complex and L1 in CD₂Cl₂. (b) High resolution electrospray ionization mass spectrometry of the isolated complex. (c) Comparison of catalytic efficiency of fresh Cu(OTf)₂ and L1 with the recycled complex.

The complex was isolated by removal of the solvent after reaction and extraction with CH₂Cl₂ (5 mL) and H₂O (5 mL). The organic phase was washed with H₂O (3×5 mL) and concentrated under vacuum. The complex was purified by column chromatography (mobile phase: 10% methanol (volume ratio) in CH₂Cl₂).

We also added TOFs to Fig. S1.

Reviewer #2 (Remarks to the Author):

The manuscript “Direct Conversion of Lignin to Functionalized Diaryl Ethers via Oxidative Cross-coupling” by Prof. Paul J. Dyson and Mingyang Liu describes a novel strategy of phenolic monomers’ extraction from lignin. In this study, the authors applied oxidative copper-catalyzed cross-coupling reaction of beech lignin with phenylboronic acids affording functionalized diaryl ethers in a simple and highly efficient manner. The authors proposed an effective procedure of lignin valorization consisting of two steps: oxidative coupling and methylation of the acid. The reaction conditions were optimized, including copper (II) source, base, ligand, solvent, catalyst loading, oxygen pressure, time, and temperature. As one of the most significant advances established by the authors, this protocol enables an efficient generation of various ethers from beech, poplar, and pine lignin in near-theoretical maximum yields with suppression of repolymerization and dearylation side reactions of the reactive phenol intermediates. The authors provided a detailed analysis of the HSQC spectra of the initial lignin polymer and final reaction mixtures. The authors found a correlation between the ratio of the linkages in lignin of three different types and the structures of the observed products. A scope of arylboronic acids with various substituents (EDG and EWG) was performed. Finally, several convincing mechanistic studies were elaborated: cleavage of C-C and C-O bonds and Cham-Lam coupling with simple substrates c-i, k under standard conditions. For the technical part, the supporting information includes the perfectly elaborated calculation of the ethers’ yield based on starting lignin, as well as the linkages content calculation and calibration by the HSQC NMR spectroscopy technique. The spectral data corresponds to the proposed structures and is well-delivered. The citations are perfectly balanced.

Overall, I believe this impactful discovery dramatically pushed the boundaries of the lignin depolymerization topic. Undoubtedly, these advances in the direct conversion of lignin will interest a broad scientific audience. Thus, I fully support the publication of this manuscript in *Nat. Commun.* with minor revisions and four additional experiments outlined below.

We thank you for the positive remarks.

1. Have you tried to subject lignocellulosic biomass, e.g., wood sawdust, directly to your reaction conditions? Is it possible to isolate any quantity of the pure desired product 1b from lignocellulose with the proposed method?

As suggested, we tested beech wood sawdust, but although the yield of products is low we added the following to the manuscript and SI:

Raw beech wood sawdust was tested using the standard reaction conditions with diaryl ether products obtained in only 10% yield (Fig. S11).

Figure S11. Application of beech wood sawdust as the lignin source. Reaction conditions: (1) Beech wood (100 mg), 4-chlorophenylboronic acid (1.5 equiv.), Cu(OTf)₂ (30 mol%), L1 (30 mol%), K₂CO₃ (4 equiv.), biphenyl (0.02 mmol, internal standard), DMSO (2 mL), O₂ (3 atm), 140 °C, 6 h. (2) K₂CO₃ (3 equiv.), MeI (10 equiv.), 25 °C, 12 h.

2. Does coupling of beech lignin with cyclohexylboronic acid (CAS 4441-56-9) result in product 1b under the established conditions? Although there is an entry with methylboronic acid 15a, a cyclic alkylboronic acid should be tested.

We evaluated cyclohexylboronic acid and added the data to Table 1 and commented on the result in the discussion:

Alkyl boronic acids (**15a**, **16a**) do not function as coupling reagents.

Table 1. Substrate scope of the boronic acids and related coupling reagents.

Coupling reagent	Ether product	Coupling reagent	Ether product	Coupling reagent	Ether product
 1a	 1b 68%	 2a	 6b 66%	 3a	 7b 68%
 4a	 8b 71%	 5a	 9b 71%	 6a	 10b 64%
 7a	 11b 49%	 8a	 12b 50%	 9a	 13b 56%
 10a	 14b 61%	 11a	 6b 55%	 12a	 6b 55%
 13a	 6b 23%	 14a	 6b 0%	 alkyl= Me, 15a cyclohexyl, 16a	 15b 0%, 16b 0%

Reaction conditions: (1) beech lignin (40 mg), boronic acid or borate ester (1.5 equiv.), Cu(OTf)₂ (30 mol%), L1 (30 mol%), K₂CO₃ (4 equiv.), biphenyl (0.02 mmol), DMSO (2 mL), O₂ (3 atm), 140 °C, 6 h. (2) K₂CO₃ (3 equiv.), MeI (10 equiv.), 25 °C, 12 h.

3. Can you recycle copper (II) heterogeneous catalyst and reuse it? How efficient will the recycled catalyst be for beech lignin valorization to produce 1b?

We have included recycling/reuse studies (Fig. S1), see response to reviewer 1, part 4.

4. What is the product yield with 10 mol% amount of Cu(II) catalyst?

We performed the requested experiment and added the results to Table S6:

Table S6. Optimization of the amount of catalyst and co-catalyst.

Entry	Cu(OTf) ₂	L1	Yield of ether (%)	Yield of 1b (%)
1	10 mol%	10 mol%	5	4
2	15 mol%	15 mol%	15	13
3	30 mol%	30 mol%	92	68
4	60 mol%	60 mol%	84	57
5	100 mol%	100 mol%	76	44
6	30 mol%	0	41	29
7	30 mol%	15 mol%	71	55
8	30 mol%	80 mol%	65	42

Reaction conditions: (1) beech lignin (40 mg), 4-chlorophenylboronic acid (1.5 equiv.), Cu(OTf)₂ (**15-100 mol%**), L1 (**0-100 mol%**), K₂CO₃ (4 equiv.), biphenyl (0.02 mmol), DMSO (2 mL), O₂ (3 atm), 140 °C, 6 h. (2) K₂CO₃ (3 equiv.), MeI (10 equiv.), 25 °C, 12 h.

5. Although poplar lignin is a bit more efficient under your reaction conditions (95% yield of ethers, 71% of 1b), you performed a scope with beech lignin (92% yield of ethers, 68% of 1b). Is there any explanation for this choice?

The reason is not a scientific reason, but it is because beech (*Fagus sylvatica*) is much more prevalent in our region in Switzerland than poplar (Swiss National Forest Inventory NFI, www.wsl.ch). We decided not to mention this in the revised manuscript, basically we have lots of it in the lab.

6. In the manuscript, Figure 2a: the heading of the table ‘Yield of ether (%)’ should be corrected to ‘Yield of ethers b (%)’ for clarity.

We have corrected the error.

7. In the manuscript, line 63: specification should be added: “the diaryl ether products (1-4b) in 92% yield (Fig. 2a, Entry 5).

We added this specification.

8. In the manuscript, a general comment: please enhance the quality of the chemical structures in the schemes.

High-quality figures and tables have been provided.

9. In the SI, line 221: rename to Figure S14.

We have corrected the error.

REVIEWERS' COMMENTS

Reviewer #2 (Remarks to the Author):

The revised version is now acceptable.